# MMP-9 as Prognostic Marker for Brain Tumours: A Comparative Study on Serum-Derived Small Extracellular Vesicles

**DOI:** 10.3390/cancers15030712

**Published:** 2023-01-24

**Authors:** Gabriella Dobra, Edina Gyukity-Sebestyén, Mátyás Bukva, Mária Harmati, Valentina Nagy, Zoltán Szabó, Tibor Pankotai, Álmos Klekner, Krisztina Buzás

**Affiliations:** 1Laboratory of Microscopic Image Analysis and Machine Learning, Institute of Biochemistry, Biological Research Centre, Eötvös Loránd Research Network (ELKH), H-6726 Szeged, Hungary; 2Doctoral School of Interdisciplinary Medicine, Albert Szent-Györgyi Medical School, University of Szeged, H-6720 Szeged, Hungary; 3Department of Immunology, Albert Szent-Györgyi Medical School, Faculty of Science and Informatics, University of Szeged, H-6720 Szeged, Hungary; 4Department of Medical Chemistry, Albert Szent-Györgyi Medical School, University of Szeged, H-6720 Szeged, Hungary; 5Institute of Pathology, Albert Szent-Györgyi Medical School, University of Szeged, H-6720 Szeged, Hungary; 6Genome Integrity and DNA Repair Group, Hungarian Centre of Excellence for Molecular Medicine (HCEMM), University of Szeged, H-6720 Szeged, Hungary; 7Department of Neurosurgery, Faculty of Medicine, University of Debrecen, H-4032 Debrecen, Hungary

**Keywords:** liquid biopsy, small extracellular vesicles, matrix metalloproteinase-9, central nervous system diseases, brain tumour, prognostic marker, survival, glioblastoma

## Abstract

**Simple Summary:**

The invasive nature of brain tumours, particularly glioblastoma, severely limits its therapy. Matrix-metalloproteinases (MMPs), enzymes involved in the degradation of the extracellular matrix, are associated with the invasiveness of brain tumours; hence, the determination of MMPs is critical for the monitoring of cancer patients. The aim of our comparative study was to evaluate the possible additional utility of the MMP-9 level of serum-derived small extracellular vesicles (sEVs) for characterising brain tumours. We established a relationship between low MMP-9 content in sEVs and improved survival, and discovered that MMP-9 levels considerably differed between tumour types and stages, showing a positive correlation with aggressiveness. We demonstrated on a large number of samples that the high MMP-9 level of serum-sEVs may serve as a negative prognostic marker for brain tumours.

**Abstract:**

Matrix metalloproteinase-9 (MMP-9) degrades the extracellular matrix, contributes to tumour cell invasion and metastasis, and its elevated level in brain tumour tissues indicates poor prognosis. High-risk tissue biopsy can be replaced by liquid biopsy; however, the blood–brain barrier (BBB) prevents tumour-associated components from entering the peripheral blood, making the development of blood-based biomarkers challenging. Therefore, we examined the MMP-9 content of small extracellular vesicles (sEVs)—which can cross the BBB and are stable in body fluids—to characterise tumours with different invasion capacity. From four patient groups (glioblastoma multiforme, brain metastases of lung cancer, meningioma, and lumbar disc herniation as controls), 222 serum-derived sEV samples were evaluated. After isolating and characterising sEVs, their MMP-9 content was measured by ELISA and assessed statistically (correlation, paired *t*-test, Welch’s test, ANOVA, ROC). We found that the MMP-9 content of sEVs is independent of gender and age, but is affected by surgical intervention, treatment, and recurrence. We found a relation between low MMP-9 level in sEVs (<28 ppm) and improved survival (8-month advantage) of glioblastoma patients, and MMP-9 levels showed a positive correlation with aggressiveness. These findings suggest that vesicular MMP-9 level might be a useful prognostic marker for brain tumours.

## 1. Introduction

In tumour diagnostics, computed tomography (CT) and magnetic resonance imaging (MRI) scans are used to determine disease status, as well as to evaluate the response to treatments [1]; however, these techniques have well-known limitations [2]. MRI, for instance, can only detect tumour masses of sufficient size, and the treatment-related changes may overlap with residual or recurrent tumours [3]. Furthermore, it is also difficult to distinguish between central nervous system (CNS) malignancies such as glioblastoma multiforme and brain metastases [4], and MRI has limited applicability to identify long-term recurrence [5].

Further standard methods for profiling tumours require obtaining tumour samples through invasive surgical procedures. In addition to carrying a high risk of complications, the limitations of such invasive procedures include difficulty in acquiring tumour samples with high heterogeneity or in inaccessible positions and taking multiple biopsies in the occurrence of metastases, as well as monitoring tumour response or relapse. Thus, neuro-oncological research aims to discover novel methods suitable for monitoring CNS tumours in clinical practice [6].

Considering the challenges associated with traditional biopsies, recent oncology research has turned its focus from analysing whole tissues to analysing various biological fluids for tumour-derived components [7,8,9]; this technique is known as liquid biopsy (LB). LBs have drastically revolutionised the field of clinical oncology, offering ease in tumour sampling, continuous monitoring by repeated sampling, devising personalised therapeutic regimens, and screening for therapeutic resistance [10]. Tumour tissues release proteins, nucleic acids, circulating tumour cells, platelets, and tumour-derived extracellular vesicles into the bloodstream [11]; all these may serve as cancer biomarkers accessible via LB.

Extracellular vesicles (EVs) are small, lipid bilayer-enclosed vesicles released by both normal and neoplastic cells into the extracellular space [12] which enter into the circulation [13,14,15]. EVs are promising cancer biomarkers accessible through LB because they are cell-secreted, nano-sized, and stable in all body fluids [16,17]. Evs contain a sample of cytosolic milieu of the donor cells, including an abundance of DNA, RNA, proteins, and other analytes [18,19,20,21], while externally they also resemble their cell of origin [22]. Mounting evidence indicates that EVs contain a wealth of information that can be used to improve cancer diagnosis and prognostic evaluation [23,24,25]. The majority of the studies mainly focused on nucleic acids [26], but EV proteins are also receiving more and more attention in cancer diagnostics [27,28].

Finding blood–based biomarkers for CNS tumour monitoring is more challenging, as the blood–brain barrier (BBB) may prevent the release of tumour-related biomarkers into peripheral blood. The available evidence supports that tumour-derived EVs can cross the BBB [29,30]. However, currently no clinically relevant EV biomarkers are accepted for the monitoring of CNS tumours. 

In a previous study, we analysed the protein content of whole serum as well as serum-derived small extracellular vesicles (sEVs)—EVs with a diameter of 30–200 nm size [31], theoretically called exosomes—of 96 patients suffering from CNS diseases. Comparative proteomic analysis by liquid chromatography–mass spectrometry (LC-MS) revealed that samples enriched in sEVs can provide an amplified source of relevant information; thus, sEVs may be more suitable than whole sera for separating patients with distinct CNS diseases using a protein panel [32]. Among the proteins of the determined panel, the matrix metalloproteinase-9 (MMP-9) was the most effective candidate in the separation of the patient groups. 

Matrix metalloproteinases (MMPs) are zinc-dependent endopeptidases that play an essential role in the physiology of cells by degrading the extracellular matrix (ECM) [33]. It has been found that ECM has an important role also in cancer progression [34]. Some ECM proteins, such as fibronectin, thrombospondin-1, laminin, and osteopontin influence the phenotype of the tumour by affecting cell migration or angiogenesis. Interaction between cancer cells and ECM components is essential for cell transformation as well as carcinogenesis [35]. In addition to the ECM degradation, MMPs have multiple biological functions in all stages of cancer, from initiation to the development of metastases [36,37,38]. Although they are associated with cancer cell survival and tumour spread, MMPs are synthetised by cancer cells in a very small amount. By secreting interleukin, interferon, growth factors, and extracellular MMP inductors, cancer cells stimulate surrounding host cells to produce MMPs in a paracrine manner [39]. MMPs secreted by normal cells can be bound to the cancer cell surface and can be utilised by the tumour cells [34].

Matrix metalloproteinase-9 (also known as 92 kDa type IV collagenase or gelatinase B) is one of the most complex forms of MMPs [33]. The role of MMP-9 in tumour tissue invasion and metastasis formation was already described in 1990 [40], and the elevated levels of MMP-9 in human brain tumours was also been reported in the 1990’s [41]. Rao et al. also showed that the expression of MMP-9 is significantly upregulated in highly malignant gliomas and correlates with the progression, suggesting a role for MMP-9 in promoting the observed invasiveness [42].

In recent decades, several studies reported the upregulation of MMP-9 in the source of tumour tissue, which provides the opportunity to identify MMP-9 in the serum as well. In the case of some peripheral tumours, elevated MMP-9 levels in plasma were detected and associated with cancer development, invasion, and poorer survival [43]; however, according to other research, MMP-9 in blood samples cannot be considered useful in estimating the invasive capacity of cancer [44]. In the case of brain tumours, specifically for glioblastoma, serum and plasma MMP-9 content analyses yielded conflicting results [45,46,47,48].

Based on the findings of our previous study [32], we hypothesised that the contradictions in the literature could be resolved by subjecting the serum-derived sEVs instead of whole serum to the MMP-9 analysis, since EVs are stable in the blood and can cross the BBB, thereby allowing the identification of the tumour-specific distinctive biomolecules using a non-invasive, simple blood test.

For this purpose, in this study, 222 serum samples were collected from four patient groups according to the criteria of the National Ethical Committee, and MMP-9 analysis was performed on serum-derived sEV samples using enzyme-linked immunosorbent assay (ELISA). According to our knowledge, this was the first study that investigated the MMP-9 level of serum sEVs in human brain tumours. The large sample size allowed us to demonstrate the factors influencing the MMP-9 level (age, gender, surgical resection, recurrence, therapy), and was suitable for determining a measurable, significant difference in the sEV MMP-9 level associated with different diseases and patient survival outcomes. Calculating the MMP-9 concentrations of serum-derived sEVs enabled us to establish the cut-off values and the specificity and sensitivity of the analysis required in clinical practice.

## 2. Materials and Methods

### 2.1. Patients

Blood samples of 222 patients treated at the Department of Neurosurgery, of the University of Debrecen were analysed. Samples were obtained from patients with glioblastoma multiforme (GBM), meningioma (M), and brain metastasis originating from non-small cell lung cancer (BM). Control samples (CTRL) were collected from patients with lumbar disc herniation without evidence of cancer. In addition to the four main groups, subgroups were also distinguished based on histopathology, sampling time (prior or after to surgery), grading, recurrence, and treatment (Table 1). Detailed patient characteristics can be found in Appendix A.

Blood samples were collected, processed to serum, and stored by the Neurosurgical Brain Tumour and Tissue Bank of Debrecen according to the criteria of the National Research Ethics Committee. An informed consent form was signed by each patient; the study was conducted in accordance with the Declaration of Helsinki. This study was carried out according to two ethical approvals, namely 51450-2/2015/EKU (0411/15), Medical Research Council, Scientific and Research Ethics Committee, Budapest, 30 October, 2015 and 121/2019-SZTE, University of Szeged, Human Investigation Review Board, Albert Szent-Györgyi Clinical Centre, Szeged, 19 July 2019.

### 2.2. Preparation of Serum Samples

Blood samples were collected into BD Vacutainer SST II Advance Tubes, allowed to clot for at least 1 h at room temperature, and centrifuged for 20 min at 3000× *g*, 10 °C. The serum samples were stored at −80 °C until further processing.

### 2.3. sEV Isolation and Characterization

sEVs were isolated from the sera via differential centrifugation as described previously [32]. Briefly, after thawing on ice, sera were centrifuged for 30 min at 10,000× *g*, 4 °C, then for 70 min at 110,000 g, 4 °C, using a fixed-angle rotor (T-1270, Thermo Fisher Scientific, Waltham, MA, USA). The pellet was resuspended in particle-free Dulbecco’s phosphate-buffered saline (DPBS, Lonza Group Ltd., Basel, Switzerland) and stored at −80 °C until further processing. This sEV isolation protocol served to reach intermediate recovery and intermediate specificity according to the guideline ‘Minimal Information for Studies of Extracellular Vesicles 2018’ (MISEV2018) [31].

Following the main suggestions and requirements included in MISEV2018, sEVs were characterised by transmission electron microscopy (TEM), Western blot analysis (WB), and nanoparticle tracking analysis (NTA). The TEM and WB measurements were performed on a representative sample of each main group.

In order to examine sEV morphology, TEM analysis was performed using a Tecnai G2 20 X-Twin type instrument (FEI, Hillsboro, OR, USA), operating at an acceleration voltage of 200 kV. The samples were dropped on a grid (carbon film with 200 mesh copper grids (CF200-Cu, Electron Microscopy Sciences, Hatfield, PA, USA)) and dried without staining or other fixation procedure.

Confirming the presence of sEVs, CD81, CD5L, and calnexin markers were presented by Western blot analyses using NuPAGE reagents and an XCell SureLock Mini-Cell System (Thermo Fisher Scientific, Waltham, MA, USA) according to the manufacturer’s protocols. The protein content of sEV samples was determined using a Pierce BCA Protein assay kit (Thermo Fisher Scientific, Waltham, MA, USA) and a benchtop microplate reader (Multiskan RC, Thermo Labsystems, Waltham, MA, USA) according to the manufacturer’s instructions. For detection of the sEV markers in the four main groups, rabbit anti-human Alix (1:1000), rabbit anti-human CD5L (1:2000), and rabbit anti-human Calnexin (1:10,000) primary antibodies (all from Sigma-Aldrich, St. Louis, MO, USA), and HRP-conjugated anti-rabbit IgG (1:1000, R&D Systems, Minneapolis, MN, USA) secondary antibody were used. THP-1 cell line (ATCC, Manassas, VA, USA) lysate was used for the positive control for Calnexin.

Determining the nanoparticle size distribution and concentration of the tested samples, sEVs were diluted in particle free DPBS and analysed using a NanoSight NS300 instrument with 532 nm laser (Malvern Panalytical Ltd., Malvern, UK). Six videos of 60 s were recorded for each sample under constant settings (Camera level: 15; Threshold: 4, 25 °C; 60–80 particles/frame).

### 2.4. MMP-9 Analysis by LC-MS

The LC-MS analysis was performed in a previous study [32] based on the serum sEV samples of 96 from the total of 222 patients. Samples presented the four main groups, namely GBM, BM, M, and CTRL. Each group contained 24 individuals with mixed ages and genders; six-sample-pools were created from the individuals, allowing four parallel samples to be tested per group. Blood samples were collected one day prior to neurosurgical procedure in each tumour case. None of the patients received radio- or chemotherapy before tumour resection.

The detailed methodology of LC-MS measurements was described in the named article [32]. Briefly, for ’in solution’ digestion, individual samples containing 20 µg protein were diluted to 26 µL and kept at 60 °C for 30 min to unfold and reduce the proteins, and then kept at RT for 30 min to alkylate the proteins. The samples were digested overnight at 37 °C with trypsin, then stopped by formic acid. The separation of the digested samples was carried out on a nanoAcquity UPLC (Waters, Milford, MA, USA), using Waters ACQUITY UPLC M-Class Peptide C18 column. The LC was coupled to a high-resolution Q Exactive Plus quadrupole-orbitrap hybrid MS (Thermo Scientific, Waltham, MA, USA). The quantitative measurements of digested individual samples were performed in DIA mode, and the analysis was conducted in Encyclopedia 0.81. A comprehensive spectral library of 10,000 human proteins was used for peptide identification.

### 2.5. MMP-9 Analysis by ELISA

To accurately measure MMP-9 content of sEVs, LEGEND MAX Human MMP-9 ELISA Kit (Biolegend, San Diego, CA, USA) was used according to the manufacturer’s protocol. sEV isolates of 222 serum samples were measured individually, and the vesicles were disrupted in a detergent-free manner by five repeated freeze–thaw cycles to expose the entire protein content. As we examined sEV isolates, potential interferences—observed in clinical immunoassays of sera [49]—are avoided. The assay procedures are briefly summarised below.

A standard curve was applied for each assay, and all samples were run in duplicate on separate plates. In the first step, 50μL assay buffer and 50 uL standard dilutions or 12× diluted samples were added to the appropriate wells, and the plates were sealed and incubated at room temperature for 2 h while shaking at 200 rpm. After the incubation, plates were washed four times with 1× wash buffer. The first step was followed by three more incubation steps, namely 100 μL of human MMP-9 detection antibody solution, 100 μL of Avidin-HRP solution, and 100 μL of substrate solution D were added to each well and incubated at room temperature for 1 h, 30 min, and 15 min, respectively. The plates were washed four times with 1× wash buffer between each incubation procedure. The last step was performed in the dark, and then the reaction was stopped by adding 100 μL of stop solution; the absorbance was immediately read at 450 nm and 570 nm on a benchtop microplate reader (Multiskan RC, Thermo Labsystems, Waltham, MA, USA).

### 2.6. Statistical Analyses

Before conducting any statistical tests, the MMP-9 level was normalised to the protein content of serum sEVs, meaning that MMP-9 concentration (ng/mL) was divided by protein concentration of the EV-enriched isolates (ng/mL) for every sample. The results of all the analyses are notated in parts-per-million (ppm), describing the individual values of MMP-9 in patients.

Outliers from the analysed groups were always excluded using the ROUT (robust regression followed by outlier identification) method with the Q parameter set to 1%. Then we examined the assumptions of normal distribution utilising the Shapiro–Wilk test and the homogeneity of variances by using the F test (comparing two groups) and the Brown–Forsythe test (comparing more than two groups).

The MMP-9 ppm concentrations of independent groups and matched samples were compared with Welch’s test and the paired *t*-test, respectively. One-way ANOVA was used to examine differences between more than two groups.

To determine the relationship between continuous variables, linear regression analyses were conducted. To improve linearity in regression, skewed data were logarithmised.

The diagnostic potential of the MMP-9 ppm was evaluated using receiver operating characteristic (ROC) analyses. Kaplan–Meier analyses with log-rank tests were used to compare the overall survival rates of different groups.

The collected data about the MMP-9 content of sEVs were analysed statistically using GraphPad Prism 8.3.4.

### 2.7. Data Availability

All datasets generated during the current study are available from the corresponding author upon reasonable request. All relevant data of our experiments had been submitted to the EV-TRACK [50] knowledgebase (EV-TRACK ID: EV230005).

## 3. Results

### 3.1. The sEVs’ MMP-9 Content Measured by ELISA Is Consistent with the Previous LC-MS Results

Our main goal was to compare the MMP-9 content of serum-derived small extracellular vesicle (sEV) samples from 222 patients with different CNS diseases. The four main investigated groups were glioblastoma (GBM), brain metastasis originated from non-small cell lung cancer (BM), meningioma (M), and lumbar disc hernia patients as controls (CTRL).

In a previous study, following the isolation and characterisation of sEV samples, LC-MS analysis was performed on 96 sEV samples. Each of the four groups (GBM, BM, M, and CTRL) contained 24 individuals, and six sample pools were created from the individuals allowing four parallel samples to be tested per group. As a result of the analysis, a 17 membered protein panel was constructed from the identified proteins which were able to separate the four groups with 100% efficacy [32]. In this study, we investigated which of the 17 sEV proteins were the most suitable for distinguishing patients, and found MMP-9 to be the most significant (*p* = 0.0065, multi ROC AUC = 0.86).

Following the candidate selection, in the current study, the MMP-9 content of the 96 sEV samples was measured by ELISA, and the MMP-9 intensities (from LC-MS) were compared with the averaged MMP-9 concentrations (from ELISA) of the six-sample-pools to ascertain that ELISA is also capable of distinguishing the groups. The correlation analysis confirmed that the MMP-9 concentrations measured by ELISA were highly comparable to LC-MS measurements (*r* = 0.7264; *p* = 0.0022) (Figure 1).

Therefore, we aimed to perform ELISA measurements on the larger cohort; the MMP-9 concentrations and sEV characteristics can be found in Appendix A and Appendix A, respectively. The large sample size allows us to investigate factors influencing the MMP-9 level, and is suitable for examining whether there is a measurable, significant difference in the sEV MMP-9 level associated with different diseases and patient survival outcome. Measuring the MMP-9 concentrations of serum EVs instead of MMP-9 intensities enables the determination of the cut-off values required in the clinic, as well as the specificity and sensitivity of the test.

### 3.2. Several Factors Might Influence the MMP-9 Level of the Serum-Derived sEVs

Performing comparative studies between different tumours requires examining what factors (e.g., age, gender, surgical resection, recurrence, and therapy) may influence MMP-9 levels of serum-derived sEVs (Figure 2).

The MMP-9 level was normalised to the total protein content of serum sEVs, meaning that MMP-9 concentration (ng/mL) was divided by protein concentration of the EV-enriched isolates (ng/mL) for every sample. The results of all the analyses are notated in parts-per-million (ppm) describing the individual values of MMP-9 in patients (Appendix A).

Age and gender analyses were carried out in the control group, eliminating the possible MMP-9-modifying effect of any tumour disease. Correlation analysis revealed that there is no distinct relationship between age and MMP-9 levels (Figure 2a). The MMP-9 level of serum-derived extracellular vesicles shows no significant difference between male and female patients, as determined by Welch’s test (Figure 2b). Based on these findings, further data analyses were conducted regardless of the age or gender of the patients.

Examining the effects of surgical resection on the MMP-9 level of sEVs, paired *t*-tests were completed in the GBM and BM groups (Figure 2c,d, respectively). The MMP-9 levels were similar (*p* = 0.1843, fold change = 15%) before and after the surgical resection in the case of primary GBM patients, while BM samples showed marked increase (*p* = 0.0065, fold change = 209%) after resection.

Further examination on preoperative GBM samples found a distinct difference between the original tumour and the recurrence based on MMP-9 levels of serum sEVs (Figure 2e), and the recurrence showed a lower level of MMP-9 on average (*p* < 0.0001).

Determining the influence of the administered therapy, MMP-9 levels were also compared based on whether or not GBM patients had received treatment at the time of sampling. Our result (*p* < 0.0001) indicates that therapy might decrease the MMP-9 levels of the circulating sEVs (Figure 2f).

In addition to the paired *t*-test shown in Figure 2c,d, unpaired *t*-tests (Welch’s tests) were performed on all primary and secondary GBM samples, as well as on all BM samples to determine the effect of surgical resection on a broader cohort (Appendix A). First, all the preoperative and postoperative samples were evaluated; then, subsequent exclusions were made for individuals who had relapsed or received therapy. The unpaired *t*-tests on primary GBM and BM samples (Appendix A) yielded the same results as the paired *t*-tests (Figure 2c,d); however, for the secondary GBM samples, the significant differences were erased when the recurrent or treated samples were excluded from the statistical analysis (Appendix A).

Based on our findings, we can conclude that in addition to the disease types, surgical resection, recurrence, and therapy might influence the MMP-9 level of the serum-derived sEVs. Due to these findings, all subsequent analyses were conducted exclusively on samples obtained prior to surgical resection and therapy administration.

### 3.3. MMP-9 Level of Serum sEVs Differs in Various CNS Tumours Showing a Positive Correlation with Tumour Aggressiveness

Further statistical analyses were performed in order to identify if there is a difference between the serum sEV MMP-9 levels of the patient groups (Figure 3).

As a first step, Welch’s test was used to compare the control group with all the tumour patients (Figure 3a). Based on ROC analysis, the two groups were significantly distinguishable (*p* = 0.0002) using a cut-off point of 16 MMP-9 ppm with 74% sensitivity, 61% specificity, and an AUC of 0.70 (Figure 3b).

The tumour patients then were divided into M, GBM, and BM, and we found that the differences remained significant between controls and malignant tumours, as well as between the benign and malignant tumours (Figure 3c). Separate comparison resulted in increased specificity, sensitivity values, and AUC scores in the case of control-malignant tumour comparisons at the expense of control-benign comparisons (Figure 3d).

In the last step, the patients were divided into further subgroups based on histopathology. In the subgroup analysis, primary and secondary GBM, patients with grade I and II meningiomas, and brain metastases from patients with adenocarcinoma and carcinoma planocellulare were distinguished. The detailed results are presented in Appendix A. The comparisons of the patient groups showing significant differences in MMP-9 levels resulted in AUC scores up to 0.77.

Our data indicate that patients suffering from malignant, but not benign brain tumours can be distinguished from CTRL patients based on the MMP-9 level of serum sEVs, and that the MMP-9 level of serum sEVs differs between CNS tumour types, showing a positive correlation with tumour aggressiveness.

### 3.4. The sEVs’ MMP-9 Level Might Be a Prognostic Marker for Overall Survival in GBM Patients

After analysing the influencing factors, we aimed to determine whether MMP-9 levels correlate with disease progression/patient survival. To assess the prognostic value of MMP-9 levels in serum sEVs, we analysed the preoperative serum samples from 27 GBM patients. Patients in this study were not administered by any treatment at the time of sampling (Figure 4).

To reveal the prognostic value of sEVs’ MMP-9 level on survival, subjects were divided into three groups based on their survival time (short-, medium- and long-term) using 0–2, 3–8, 10–23 months as the cut-offs (Figure 4a). Long-term survival was found to be associated with a significantly lower MMP-9 level compared to the MMP-9 levels of the other two groups (Figure 4b–d). These differences represent a high specificity and sensitivity of 80–89% at a threshold of 28 MMP-9 ppm, and AUC values of 0.83–0.87 in ROC analyses allow efficient distinction of these survival groups.

These results suggest that there should be a correlation between the MMP-9 level of serum sEVs and overall survival (OS), so we performed an analysis where we observed that lower survival time was associated with higher MMP-9 level (Figure 4e). To determine the extent of the influencing effect of MMP-9 levels on survival, patients were separated into two groups, with the previously established threshold of 28 ppm. Based on the Kaplan–Meier chart, patients with low MMP-9 level (<28 ppm) presented with a significant OS benefit (HR 2.401, 95%CI 1.095 to 5.261, *p* = 0.0063), which represents an eight-month increase in the median OS (Figure 4f).

The probability of survival also decreased with age; therefore, we repeated the examination with a cut-off of 65 years. According to the Kaplan–Meier chart, patients under the age of 65 had a five-month increase (HR 2.037, 95%CI 0.8524 to 4.869, *p* = 0.0340) in median survival (Figure 4c). It is crucial to note that age and MMP-9 level are independent (see Figure 2a); we can conclude that the MMP-9 level of serum sEVs may influence survival regardless of age.

To summarise the main findings, analysis of samples taken prior to surgical resection and the administration of therapy revealed a negative correlation between higher MMP-9 levels and the survival, and long-term (10–23 months) survival found to be associated with low MMP-9 level (<28 ppm). Our results support that the high level of MMP-9 in serum-derived sEVs might be a negative prognostic marker of the probability of survival in GBM patients.

## 4. Discussion

In recent years, oncology research has turned its focus from analysing whole tumour tissues to analysing various biological fluids for tumour-derived components [7,9]. Although several attempts have been made to identify GBM-specific biomarkers in serum [11,51,52], to the best of our knowledge, serum/plasma biomarkers are not routinely used for clinical monitoring of glioblastoma. The lack of reliable non-invasive serum-based biomarkers for GBM can be explained by several reasons, including (1) the extremely high complexity of tumour tissues (hence the name glioblastoma multiforme), (2) the barrier function of BBB that prevents the ‘tumour information’ from entering the circulation, and (3) the presence of abundant molecules that can ‘mask’ the potential biomarker candidates.

Using EV samples instead of whole sera, it is possible to amplify the signals released by brain tumours into the circulation [32]. Extracellular vesicles are stable in the blood [12], and can cross the BBB [29,30]; moreover, EVs may contain tumour-related molecules in higher concentrations and are accompanied by less contaminating molecules that may bias the analytical findings. These advantageous properties of EVs increase the possibility of identifying molecules that provide valuable information regarding tumours.

Due to the priority of living-cell secretion, large amounts and stable circulation compared to circulating tumour cells and ctDNA, exosome-based liquid biopsy has been tested in clinical trials and several of them have been approved and reached the market. In 2016, Exosome Diagnostics proposed the first exosome-based liquid biopsy in the world [22], ExoDx™ Lung (ALK), for the isolation and analysis of exosomal RNA from blood samples. In addition, the ExoDx Prostate IntelliScore (EPI) has been certified by FDA [53], and 26% of unnecessary needle biopsies were avoided with an appropriate threshold [54]. High sensitivity was achieved in prospective studies; therefore, they could be used to assist in the early diagnosis of cancer and eliminate unnecessary prostate biopsy [55].

According to Lihares et al., screening for gliomas has no clinical relevance at this time [48]. This is due to the low incidence, absence of sensitive biomarkers in plasma, and the observation that gliomas can develop apparently de novo in a matter of weeks or months. However, a non-invasive biomarker that can be used to estimate tumour state and patient survival would be of great assistance to physicians.

Based on these considerations, we sought to determine whether the MMP-9 (an endopeptidase involved in ECM degradation, thus having a role in tumour invasion) level of serum-derived sEVs can provide information about patient survival. ELISA was used to measure the MMP-9 content of 222 sEV samples isolated from whole serum for this purpose. Serum samples were collected from patients with the most common malignant, benign, and metastatic brain tumours, namely glioblastoma multiforme, meningioma [56], and brain metastasis of non-small cell lung cancer [57], and from control patients with lumbar disc herniation.

Following the measurements, comparative analyses on the MMP-9 level of sEVs were conducted. We determined that surgical resection, recurrence, and therapy can modify the MMP-9 level, but neither age nor gender can. We also detected statistically significant differences in the sEVs’ MMP-9 levels associated with various diseases, as well as a correlation between low MMP-9 level and longer survival time, which highlighted the prognostic value of vesicular MMP-9 level. However, functional studies, including MMP-9 activity measurements, would be needed to explore the causal relationship between tumour progression and the MMP-9 level of serum sEVs.

We hypothesise that the higher MMP-9 level of sEVs observed in patients with a poor prognosis is due to the tumour itself. However, there is a debate surrounding the source of the tumour-specific EVs in the blood. Osti et al., as a confirmation of the tumour origin of extracellular vesicles, employed an orthotopic transplantation of tumour-initiating cells (TICs) isolated from human GBM. GBM TICs expressing the exosomal marker CD9 coupled with the GFP protein were produced using lentiviral transduction. Because GBM-derived extracellular vesicles express CD9 on their surface, they were distinguished from natural GFP-negative mouse extracellular vesicles by following the GFP signal. According to their data, nearly half of the extracellular vesicles in the peripheral circulation of transplanted mice were tumour-derived [58]. However, using EV capture and staining techniques that allow differentiation of host cell and GBM-derived EVs, Fraser et al. demonstrated that tumoral EVs often present less than 10% of all EVs with extensive heterogeneity in tumour marker expression in GBM patient plasma [59]. The proportion of tumour-derived sEVs in the blood is a very interesting issue, although its clarification is beyond the scope of our study.

To the best of our knowledge, this study was the first one that analyses the MMP-9 content of serum sEVs; nevertheless, a few serum MMP-9 investigations have been published with contradictory findings. Hormingo et al. found that YKL-40 and MMP-9 could be monitored in patients’ serum and help confirm the absence of active disease in GBM [45]. However, after five years, the same group (Iwamoto et al.) conducted a longitudinal prospective study of MMP-9 as a serum marker in gliomas, and the larger cohort could not confirm the previous findings. Serum MMP-9 showed no utility in determining glioma disease status and was not a clinically relevant prognostic marker of survival. There was no statistically significant correlation between serum MMP-9 levels and radiographic disease status. Longitudinal increases in MMP-9 were weakly associated with shorter survival in glioblastoma patients, but they were not independently associated with survival after adjusting for age, extent of resection, and performance status [46]. In contrast, Ricci et al. confirm that MMP-9 levels are significantly higher in high-grade glioma, in low- and high-grade meningioma samples, as well as in metastasis specimens compared to healthy individuals (*p* < 0.001) [47]. This latter research of serum MMP-9 levels is in line with our sEV-based findings.

Several technical issues may have contributed to the poor and contradictory serum/plasma results. It is important to note that the measurement of MMPs in body fluids can be influenced by the type of fluid and method of collection and storage. EDTA or heparin, for instance, can affect the baseline serum/plasma concentrations of MMP-9 [60]. To solve this issue, for instance, Ricci et al. prepared native serum using plastic tubes without coagulation accelerators, to prevent the release of gelatinases during platelet activation [47]. We used BD Vacutainer SST II Advance Tubes acril gel with only silica particles as clot activators.

Another difficulty is that plasma MMP-9 may be unstable and degrades rapidly even when stored at −80 °C [61,62]. This problem can be alleviated by using sEVs instead of serum, as the high biological stability allows long-term storage of specimens for exosome isolation and detection [63]. Diagnostic analytes, RNA, DNA, or protein cargo in exosomes maintain this informative profile’s stability during sample storage, which is essential for the development of biomarkers using retrospective samples [64]. This raises the possibility of monitoring MMP-9 not only at the protein, but also at the mRNA level.

Recently, several research groups have studied vesicular mRNA that can be isolated from body fluids [65,66,67,68]. However, the majority of MMP-9 mRNA investigations are conducted on tissue samples [69,70,71], and there are only a few studies assessing MMP-9 mRNA and MMP-9 protein in serum [72,73,74]. Although these are not comparative studies of cancer patients, they indicate that increase in MMP-9 is detectable with similar efficiency at both mRNA and protein levels. At the same time, due to the complex regulation of translation and protein degradation, there is no strict correlation between mRNA and protein abundances [75,76,77]. Thus, we assume that vesicular MMP-9 could be measured at the mRNA level as well, but its prognostic value would need further investigation.

It is also important to note that there is no gold standard method for sEV isolation, which may impede the reproducibility of experimental findings in EV research. The applied technique impacts the isolation efficacy in terms of yield, purity, and EV composition, which may interfere with downstream analyses, including the molecular cargo profiling [78,79,80,81]. Another technical issue in EV research is that storage at −80 °C and freeze–thaw cycles may affect stability of EV membranes leading to EV disruption and subsequent fusion phenomena, which may influence the EV characteristics, including biological activity [82,83,84,85], but enable the use of repeated freeze-thaw cycles to expose the entire EV cargo prior to downstream analyses (as it was done here and previous studies [86,87], or to load therapeutic cargo into vesicles [88,89]. To enhance the reliability, transparency, and reproducibility of EV studies, the ISEV published the MISEV guidelines, which discuss all technical issues and provide help to set up the experimental protocols best fit to the research question [31,90].

Prior studies have suggested that MMP-9 measurements in serum do not reliably reflect circulating MMP-9 levels and may be artificially high compared to results obtained from plasma samples [43,91,92,93], but Iwamoto et al. showed that serum and plasma MMP-9 samples were highly correlated in their large sample size, so serum, and thus serum-derived sEV samples, could be used appropriately [46].

In order to find the putative source of the MMP-9 as biomarker, Ricci et al. correlated the expression of serum MMP-9 with the expression of the same protein in the tumour tissue. To address this aim, glioma tissues were subjected to immunohistochemistry with MMP-9 antibody. They discovered that the delocalisation of the MMP-9 signal from glioma cells to endothelial cells of neoplastic blood arteries is directly associated with tumour malignancy. In line with these findings, previous immunohistochemical and in situ hybridization studies demonstrated that, in high-grade glioma, MMP-9 expression is primarily restricted to perivascular regions at the infiltrating borders of the tumour and, in the majority of cases, to endothelial cells, with a close association to tumour malignant behaviour [94,95]. To determine the origin of MMP9 in GBM, Jiguet-Jiglaire et al. analysed its expression by immunohistochemistry. According to their findings, MMP-9 staining was mainly located in the microvascular proliferation, and also in inflammatory cells and circulating intravascular cells. No staining was observed in glioblastoma cells or in the extracellular matrix [39].

The successiveness of a biomarker candidate can be determined by comparing its efficacy to that of existing clinical procedures. We were unable to correlate the sEVs’ MMP-9 level of the patients with their MRI status; nevertheless, contradictory findings exist in the literature. Hormigo et al. observed that levels of MMP-9 were higher in the serum samples of patients with high-grade glioma after surgery, while the MMP-9 concentrations were significantly lower in glioblastoma patients with no radiographic evidence of disease in comparison to the subjects with active tumour [45]. In contrast, Iwamoto et al. found that there was no statistically significant correlation between serum MMP-9 levels and radiographic disease status [46]. Moreover, in a very recent study by Jiguet-Jiglaire et al., MMP-9 did not correlate with glioblastoma tumour volume, invasion, or angiogenesis assessed by neuro-imaging [39].

In addition to disease status, other patient characteristics may influence MMP-9 levels in the patient sera. Otero-Estévez et al. found that MMP-9 was significantly correlated with gender and age when examining the serum of colorectal cancer patients [44]. In contrast, in our study on serum-derived sEVs, there was no distinct relationship between age and MMP-9 levels, and there was no significant difference between male and female patients in terms of MMP-9 level (Figure 2). Based on our findings, data analyses on serum-derived sEVs can be conducted regardless of the age or gender of the patients.

The surgical resection also can affect the extracellular vesicle concentration and the MMP-9 content of serum/plasma EVs. Osti et al. measured a significant drop (*p < 0.001*) in plasma extracellular vesicle concentration after surgery [58]. Hormigo et al. observed that levels of MMP-9 were higher in the sera samples of patients with high-grade glioma after surgery, suggesting that increases in the serum level of this protein may be associated with brain inflammation and breakdown of the blood–brain barrier, rather than be a true measure of tumour burden [45]. In our study, the MMP-9 level of serum sEVs were similar before and after the surgical resection in the case of primary GBM patients (*p* = 0.1843; *n* = 14), while BM samples showed marked increase (*p* = 0.0116; fold change = 209%) after resection. This intriguing phenomenon relies on the basis of a few BM samples (*n* = 6), and therefore should be supported by further measurements.

Determining the influence of the administered therapy, MMP-9 levels were also compared based on whether or not GBM patients had received treatment at the time of sampling. Our result (*p* < 0.0001) indicated that therapy might decrease the MMP-9 level of circulating sEVs (Figure 2f). Further examination on preoperative GBM samples found a distinct difference between the original tumour and the recurrence based on the MMP-9 level of serum sEVs (Figure 3e). Although recurrence shows a lower level of MMP-9 on average (*p* < 0.0001), the majority of these GBM patients received treatment, therefore the decrease in MMP-9 cannot be attributed solely to relapse. Tabouret et al. reported that urine protein levels of MMP-2 also decreased and were considered to be related to treatment response, although this was not confirmed in an additional patient cohort [96].

One of the main findings of our study relies on the correlation analysis of samples taken prior to surgical resection and the administration of therapy on MMP-9 level and survival. A negative correlation was revealed between higher MMP-9 levels and the survival, and the long-term (10–23-month) survival was found to be associated with a low MMP-9 level (<28 ppm). Our results support that the high level of MMP-9 in serum-derived sEVs might be a negative prognostic marker of the probability of survival in GBM patients. Osti et al. tested for possible associations between plasma extracellular vesicle concentration and patient outcomes. Patients with GBM were ranked according to their extracellular vesicle content and divided into high-content or low content groups; no significant differences in PFS or OS were found [58]. Jiguet-Jiglaire tested patients with recurrent glioblastoma prior to treatment and found decreased plasma levels of MMP-9 associated with increased OS [39].

By measuring the level of MMP-9, we may determine the efficacy of the treatment. For instance, the majority of patients with GBM will experience relapse despite surgery and standard first-line treatment consisting of radiotherapy with concurrent and adjuvant temozolomide. In certain cases, therapeutic options at the time of recurrence include surgery or reirradiation, whereas in other cases, bevacizumab is the preferred option worldwide. Jiguet-Jiglaire et al. demonstrated that low baseline plasma levels of MMP-9 were associated with a high response rate and a prolonged PFS and OS in patients with recurrent GBM treated with bevacizumab but not cytotoxic chemotherapy. In addition, they observed that MMP-9 plasma levels decreased during treatment with bevacizumab and tended to rise with disease progression. In a retrospective analysis performed in the Avaglio trial (a randomised phase III trial that compared bevacizumab versus placebo in addition to standard of care in newly diagnosed glioblastoma patients), a low plasma level of MMP-9 at baseline consistently predicted PFS and OS gain associated to bevacizumab [39].

Unmet medical need exists for biomarkers able to predict response to antiangiogenic agents. According to the ClinicalTrials.gov database, only a few studies are currently being conducted to determine the predictive value of MMP-9 level on cancer treatment response. One study aims to investigate the predictive impact of circulating MMP-2 and MMP-9 on the progression-free survival of patients with metastatic kidney cancer treated with anti-angiogenic agents (Sunitinib or Pazopanib) in comparison with two untreated cohorts (NCT03185039). Another study evidenced the role for MMP-9 in the primary or acquired resistance to bevacizumab; therefore, in their trial they use monoclonal antibody GS5745 that may overcome resistance to bevacizumab through a specific inhibition of MMP-9. The phase I study is the first step to analyse the tolerance, determine the recommended dose of the combination, and explore the impact of GS5745 on MMP-9 plasma levels and multimodal imaging in patients with recurrent glioblastoma (NCT03631836).

On the other hand, Farina and Mackay draw attention to the Janus-faced nature of MMP-9. This important molecule plays an essential role in tumour biology, from initiation/promotion to angiogenesis, dissemination, immune surveillance, and metastatic growth. However, MMP-9 also possesses antitumour activity and serves essential physiological functions [97]. In order to determine the potential therapeutic efficacy of inhibiting MMP-9 function in cancer therapy, it is necessary to develop specific inhibitors of MMP-9, to inhibit the tumour promoting function of MMP-9 instead of suppressing the anti-tumour effect. Validation of MMP-9’s predictive value in a prospective study is a crucial step toward its possible routine therapeutic application. In addition, it may be advantageous to (i) investigate the invasion- and angiogenesis-related molecules co-expressed with MMP-9, or (ii) develop strategies for inhibiting tumour-specific activators of MMP-9 instead of using direct inhibitors.

In summary, we have demonstrated that sEVs’ MMP-9 content is suitable for estimating the probability of patient survival and it allows us to obtain information about CNS tumour aggressiveness. This biomarker’s benefits include its accessibility (a simple blood test), affordability, fast detectability, ease of implementation, and reproducibility. Collectively, these advantageous characteristics would permit the application of serum sEVs’ MMP-9 for non-invasive patient monitoring.

## 5. Conclusions

Glioblastoma is the most prevalent primary brain tumour in adults, comprising 45.2% of all malignant primary CNS tumours, with a median survival time of 15 months. On average, 5.5% of patients live five years after their diagnosis [98].

The link between the survival of glioblastoma patients and their MMP-9 level of serum-derived sEVs is revealed in our article; hence, the identification of sEVs’ MMP-9 is of great importance. Our presented fast-to-perform and non-invasive method can support clinical decision making.

## Figures and Tables

**Figure 1 cancers-15-00712-f001:**
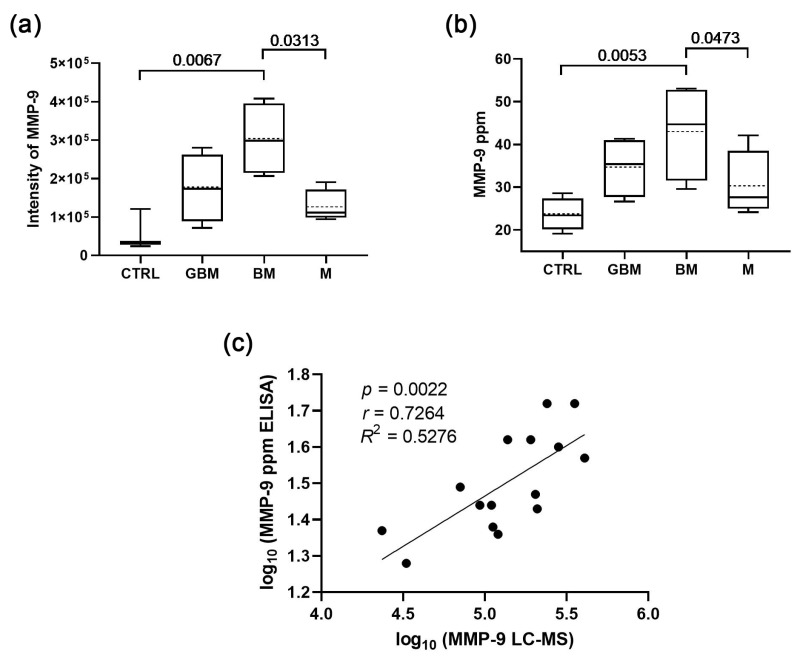
**Comparison of the sEV MMP-9 content measured by ELISA with the LC-MS results.** (**a**) MMP-9 levels of serum-derived sEVs based on LC-MS data (boxplot shows median with interquartile range, mean with dotted line, error bars range from the 5th–95th percentiles (nCTRL = 3, nGBM = 4, nBM = 4, nM = 4). (**b**) MMP-9 levels of serum-derived sEVs based on ELISA data (nCTRL = 4, nGBM = 4, nBM = 4, nM = 4). Dotted lines indicate mean values. (**c**) Correlation between the MMP-9 levels measured by LC-MS and ELISA.

**Figure 2 cancers-15-00712-f002:**
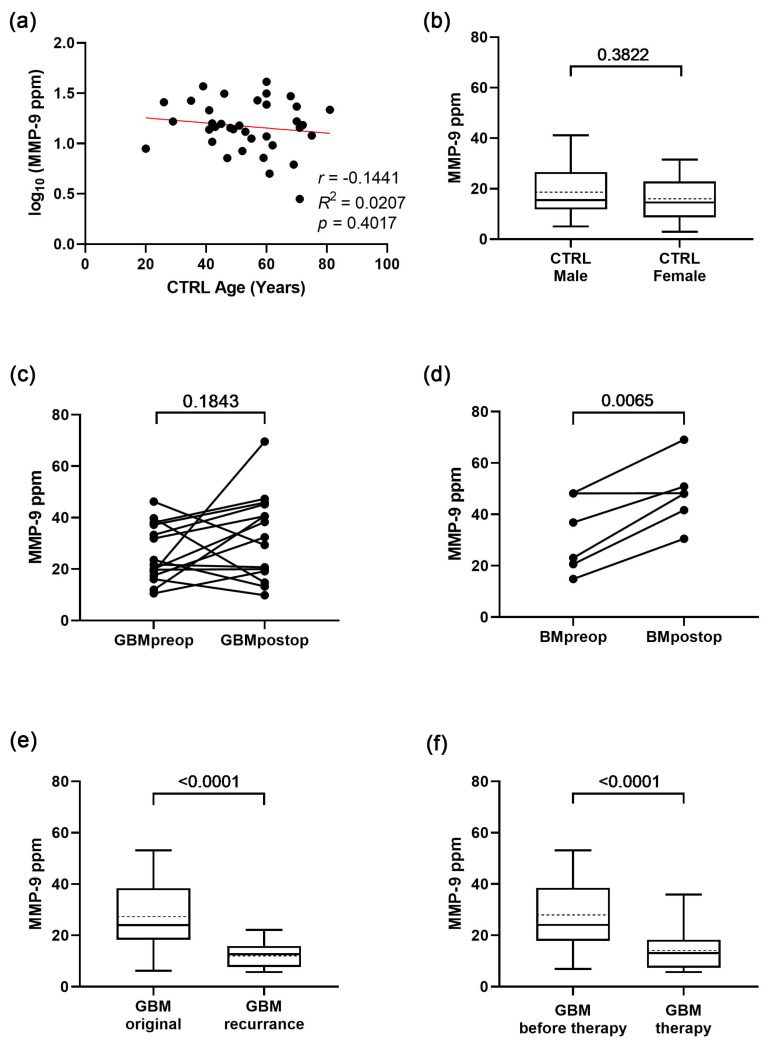
**Factors influencing MMP-9 level of serum-derived sEVs**. (**a**) Relationship between age and MMP-9 levels in the control group. (nCTRL = 36). (**b**) MMP-9 levels of the two genders in the control group (boxplots show the median with interquartile range, mean with dotted line, error bars range from the 5th–95th percentiles; nCTRL male = 16, nCTRL female = 20). (**c**,**d**) Changes in individual patient’s MMP-9 levels before and after surgical resection for the GBM and BM groups, respectively (nGBM preop-postop = 14, nBM preop-postop = 6). (**e**) MMP-9 levels of serum-derived sEVs regarding the original tumour and the recurrence in GBM group before surgical resection (nGBM original = 52, nGBM recurrence = 14). (**f**) MMP-9 levels in GBM group to treatment status (nGBM before therapy = 52, nGBM therapy = 15).

**Figure 3 cancers-15-00712-f003:**
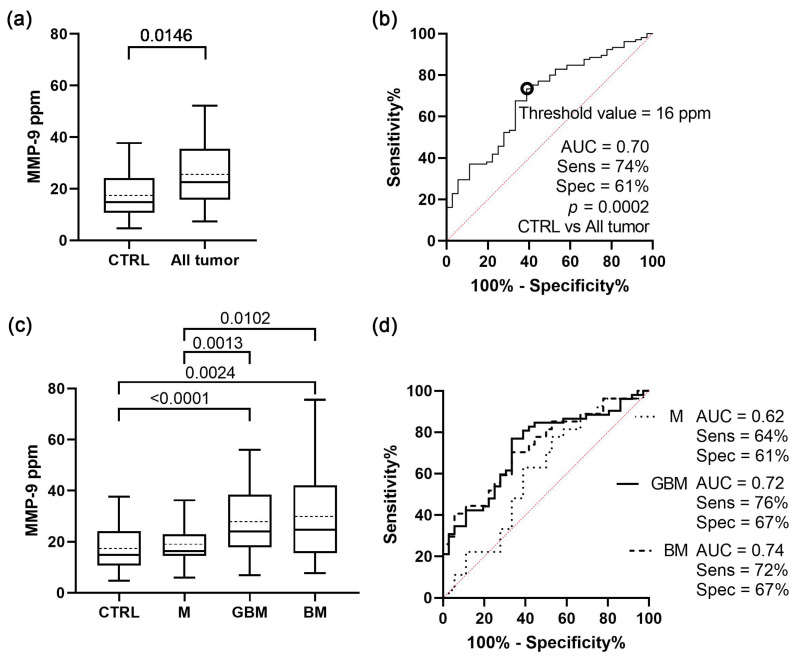
**Comparative analyses of MMP-9 level of serum sEVs on the four main groups.** (**a**) Graph shows the MMP-9 level of the control group comparing all the tumour patients (median with interquartile range, mean with dotted line, error bars range from the 5th–95th percentiles; nCTRL = 36, nAll tumour = 105). (**b**) ROC curves for exploring the differences between the MMP-9 level of the control group and all the tumour patients (nCTRL = 36, nAll tumour = 109). (**c**) Diagram shows the MMP-9 level of serum sEVs among the four patient groups (nCTRL = 36, nM = 27, nGBM = 52, nBM = 27). (**d**) ROC curves for comparing the three tumour groups to controls (nCTRL = 36, nM = 27, nGBM = 52, nBM = 27).

**Figure 4 cancers-15-00712-f004:**
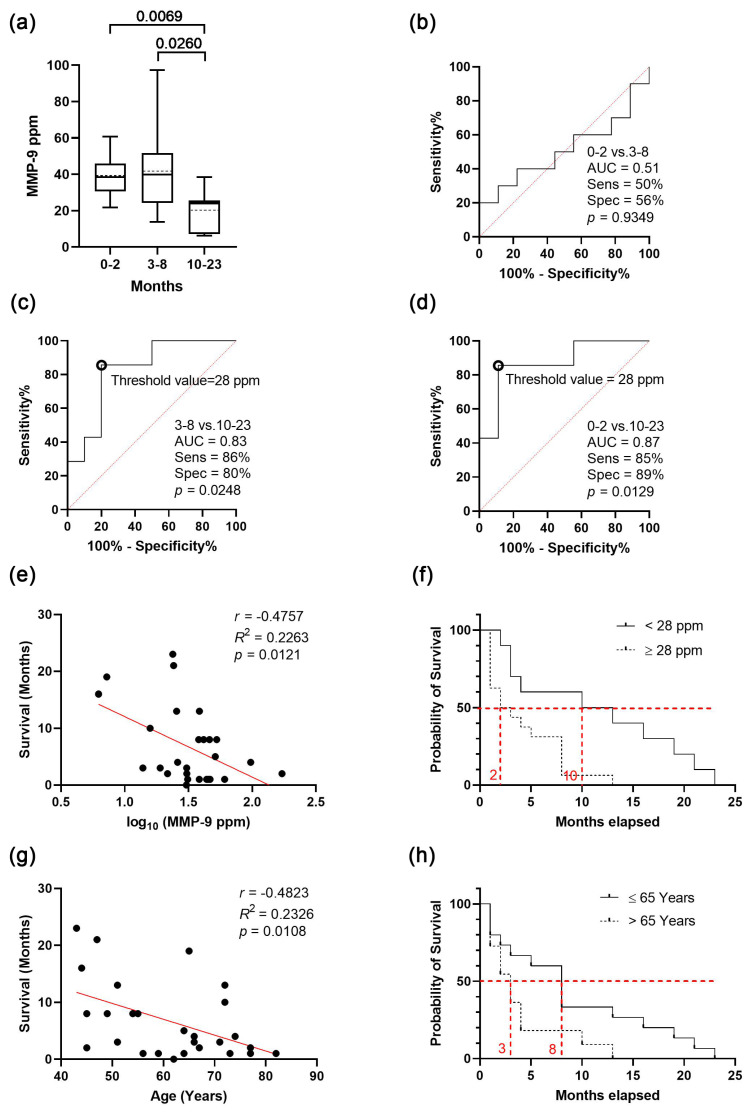
**Investigation of survival in GBM patients**. (**a**) Differences among the short-, medium- and long-term survival based on MMP-9 level of serum derived sEVs of GBM patients (median with interquartile range, mean with dotted line, error bars range from the 5th–95th percentiles, dotted lines indicate mean values). (**b**–**d**) Receiver operating characteristic (ROC) curves for comparing the short- (0–2 months), medium- (3–8 months) and long-term (10–23 months) survival groups (threshold values of the group-membership score are marked with black circles). (**e**) Correlation between MMP-9 level and survival in the GBM group (nGBM = 27). (**f**) Overall survival according to low and high (*n* < 28 ppm = 10 and *n* ≥ 28 ppm = 17) MMP-9 level. (**g**) Relationship between age and survival in the GBM group (nGBM = 27); (**h**) Overall survival according to the age (*n* ≤ 65 years = 16 and *n* > 65 years = 11).

**Table 1 cancers-15-00712-t001:** Patient cohort and participation in the statistical analyses.

Characteristics	*n* = 222	%
**Glioblastoma multiforme (GBM)**	**121**	**54%**
*secondary glioblastoma (GBMsec)*	*18*	*15%*
preoperative samples (GBMsec preop)	9	50%
postoperative samples (GBMsec postop)	9	50%
*primary glioblastoma (GBMprim)*	*103*	*85%*
preoperative samples (GBMprim preop)	69 (10) ^1^	67%
recurrence-related analysis	69	67%
original tumour	54	78%
recurrence	15	22%
therapy involvement analysis	69	67%
patients before therapy	54	78%
patients with therapy	15	22%
survival analysis	27	39%
>65 Years	11	41%
≤65 Years	16	59%
high MMP-9 level (≥28 ppm)	17	63%
low MMP-9 level (<28 ppm)	10	37%
postoperative samples (GBMprim postop)	14 **(10)** ^1^	33%
**Brain Metastasis (BM)**	**37**	**17%**
*carcinoma planocellulare (BMplano)*	*13*	*35%*
*adenocarcinoma (BMadeno)*	*24*	*65%*
preoperative samples (BMpreop)	27 (6) ^2^	73%
postoperative samples (BMpostop)	10 (6) ^2^	27%
**Meningioma (M)**	**28**	**13%**
*meningioma Grade I (M_I)*	*20*	*71%*
*meningioma Grade II (M_II)*	*8*	*29%*
**Control (CTRL)**	**36**	**16%**
*lumbar disc herniation*	*36*	*16%*
male	16	44%
female	20	56%

Percentages indicate the participation rates within each statistical analyses; ^1^ Number of GBM patients involved in paired *t*-test; ^2^ Number of BM patients involved in paired *t*-test.

## Data Availability

All datasets generated during the current study are available from the corresponding author upon reasonable request.

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
