# Peer review of "MMP-9 as Prognostic Marker for Brain Tumours: A Comparative Study on Serum-Derived Small Extracellular Vesicles"

_cancers, 2023, doi:10.3390/cancers15030712_

Round 1

Reviewer 1 Report

This is a really interesting study, proposing a new non-invasive biomarker for GBM. Some minor recommendations are listed below for amendment of the manuscript

Although the authors stated that they followed MISEV recommendations and give an EV Track entry there is no evidence for this within the manuscript itself. Please include figures on characterisation of EVs (NTA, AFM and WBs) isolated from the additional samples as a supplementary figure.

The authors describe the identification of MMP-9 protein by ELISA. How do you know if the protein is activated? Have they carried out zymography on their samples or a similar assay? The need for functional assays should be emphasised. This would also resolve the controversy surrounding the Jiguet-Jiglaire  paper mentioned on p15 regarding a lack of correlation with invasion and angiogenesis.

P6 line212-3 what does the phrase ‘the samples were unearthed by five –repeated freeze-thaw cycles’ mean?

Pg6 lines 216-219 what does the * symbol mean – times?

Figure 3 ROC values are acceptable i.e. over 0.7/70% what do you need to do to improve these – more samples/prospective trials rather than retrospective data analysis?

P15 line 500-506 the authors state sEVs are more stable which is correct but recent evidence has indicated that some degradation does occur particularly after repeated freeze thaw cycles. Please reference and mention this.

Were any angiogenesis markers co-expressed in the LC-MS data?  The link to angiogenesis and bevacizumab response is really interesting and brings me back to my comment above about the need to assay activity not just presence. This also links to their comment on the ‘Janus- faced nature of MMP-9’ perhaps the best way to modulate activity is to inhibit a context dependent activator rather than MMP-9 itself?

Author Response

Please see our answeres in the attachment.

Reviewer 2 Report

Dobra et al. investigated the potential of serum-derived sEV-associated protein, MMP-9, as a prognostic marker of glioblastoma multiforme. This is a high-quality study with a decent number of patient samples to generate statistically significant values. The discussion is thorough and has sound arguments compared with the previous findings.

In order to further improve the quality, I will suggest including a paragraph on variability in results caused by the different methods of isolation. Unfortunately, the field has not yet reached a consensus on the EV isolation method. Thus, it will merit mentioning the influence of different isolation methods on EVs' molecular profile.

Author Response

(The authors gave the same response as above.)

Reviewer 3 Report

This is a timely study aimed at testing exosomal biomarker (MMP9) from liquid biopsy derived from brain cancer patients. I have minor suggestions.

1) page 7 line 259: Non small cell lung cancer is wrongly written

2)is it possible to test mmp9 mRNA in these specimens? Would the authors expect similar results? Please add this to discussion.

3) ELISA employs biotin streptavidin based teachnology, please mention about the potential interferences

4)It will be more credible if authors include a negative control 

Author Response

(The authors gave the same response as above.)
